

# Veros v0.1 — a Fast and Versatile Ocean Simulator in Pure Python

Dion Häfner[1], René Løwe Jacobsen[1], Carsten Eden[2], Mads R.B. Kristensen[1], Markus Jochum[1], Roman Nuterman[1], and Brian Vinter[1]

[1]Niels Bohr Institute, University of Copenhagen, Copenhagen, Denmark
[2]Institut für Meereskunde, Universität Hamburg, Hamburg, Germany

**Correspondence:** Dion Häfner (mail@dionhaefner.de)

**Abstract.** A general circulation ocean model is translated from Fortran to Python. It is described how its code structure is optimized to exploit available Python utilities, remove simulation bottlenecks, and comply with modern best practices. Furthermore, support for Bohrium is added, a framework that provides a just-in-time compiler for array operations, and that supports parallel execution on both CPU and GPU targets.

For applications containing more than a million grid elements, such as a typical $1° \times 1°$ horizontal resolution ocean model, Veros is approximately half as fast as the MPI-parallelized Fortran base code on 24 CPUs, and as fast as the Fortran reference when running on a high-end GPU. By replacing the original conjugate gradient stream function solver with a solver from the pyAMG Python package, this particular subroutine outperforms the corresponding Fortran version by up to 1 order of magnitude.

The study is concluded with a simple application in which the North Atlantic wave response to a Southern Ocean wind perturbation is investigated. It is found that even in a realistic setting the phase speeds of boundary waves matched the expectations based on theory and idealized models.

*Copyright statement.* TEXT

## 1 Introduction

Numerical simulations have been used to further our understanding of the ocean circulation for more than 50 years now (e.g., Bryan, 2006), and in particular for regimes that are difficult to treat analytically, they have become irreplaceable. However, numerical representations of the ocean have their own pitfalls, and it is paramount to build trust in the numerical representation of each and every process that is thought to be relevant for the ocean circulation (e.g., Hsieh et al., 1983). The last 20 years have seen a massive increase in computing resources available to oceanographers, in contrast to human resources, which appear to be fixed. Arguably, this lead to a shift from process studies to analysis of climate model output (or from "Little Science" to "Big Science", Price de Solla, 1963). This is not necessarily a bad development, it may simply be an indication that the field has matured. However, there are still basic questions about ocean dynamics that yet remain unanswered (e.g., Marshall



and Johnson, 2013), and to tackle these questions, the scientific community requires flexible tools that are both approachable, powerful, and easy to adapt. We therefore decided to build Veros (the versatile ocean simulator).

The ocean interior is mostly adiabatic and has a long memory, easily exceeding 1000 years (e.g., Gebbie and Huybers, 2006). This requires long integration times for numerical models; experiments can well take several months in real time to
complete. Thus, ocean models are typically written to optimize the use of computing rather than human resources, using the low-level programming language Fortran. Regardless of the quality of the resulting code, the core language design and lack of abstraction in Fortran often makes it a daunting challenge to, for example, keep track of indices or global variables. Even for experienced scientists this is more than just a nuisance. As the model code becomes increasingly complex, it violates a core principle of science: reproducibility – one cannot ascertain beyond all doubt that the impact of a recently implemented physical
component is caused by new physics or, simply, a bug.

High-level programming languages like Python, MATLAB, Scala, or Julia on the other hand are usually designed with the explicit goal to improve code structure and readability. While this in itself cannot eliminate coding mistakes, a more concise, better structured code makes it easier to spot and avoid bugs altogether. In the case of Python, additional abstraction, a powerful standard library, and its immense popularity in the scientific community[1] – which has in turn created a wide range of learning
resources and a large third-party package ecosystem – lower the bar of entry for inexperienced programmers. In fact, this is one of our main motivations behind developing Veros: In our experience, a substantial amount of the duration of MSc or PhD projects is devoted to understanding, writing, and debugging legacy Fortran code. This leads to frustration and anxieties, even on the lecturers' side. With Veros, we anticipate that students and researchers can translate their physical insights rapidly into numerical experiments, thereby maintaining the high level of enthusiasm with which they entered the field.

The price to pay for these advantages is often a significantly reduced integration speed due to less aggressive compiler optimizations, additional overhead, and lack of direct memory access. However, in Veros, this performance impact turns out to be much less severe than expected, as all expensive computations are deferred to a well-performing numerical backend (NumPy or Bohrium, see Sect. 3.2 for performance comparisons).

The next section describes the challenges overcome during the translation and resulting changes in the code structure. Sect. 3
presents model validation and benchmarks, and Sect. 4 evaluates the properties of coastally trapped waves in Veros.

## 2    Implementation

At its numerical core, the present version of Veros (v0.1) is a direct translation of pyOM2 (v2.1.0), a primitive equation finite-difference ocean model with a special emphasis on energetic consistency (Eden and Olbers, 2014; Eden, 2016). PyOM2 consists of a backend written in Fortran 90 and frontends for both Fortran and Python (via f2py, Peterson, 2009). Most of the
core features of pyOM2 are available in Veros, too; they include:

---

[1]There are many attempts to rank programming languages by popularity, and Python is usually placed in the top 10 of such rankings; see e.g., IEEE Spectrum (2017), Stack Overflow (2017), TIOBE Group (2017), or PYPL (2017).





- A staggered, three-dimensional numerical grid (Arakawa C-grid, after Arakawa and Lamb (1977)), discretizing the Primitive Equations in either Cartesian or pseudo-spherical coordinates (e.g., Olbers et al., 2012). This grid is staggered in all dimensions, placing quantities on so-called T, U, V, W, and $\zeta$ cells.

- Free-slip boundary conditions for momentum, and no-normal-flow boundary conditions for tracers.

- Several different friction, advection, and diffusion schemes to choose from, such as harmonic / biharmonic lateral friction, linear / quadratic bottom friction, explicit / implicit vertical mixing, central difference / Superbee flux limiting advection schemes.

- Either the full 48-term TEOS equation of state (McDougall and Barker, 2011), or various linear and nonlinear model equations from Vallis (2006).

- Isoneutral mixing of tracers following Griffies (1998).

- Closures for mesoscale eddies (after Gent et al., 1995; Eden and Greatbatch, 2008), turbulence (Gaspar et al., 1990), and internal wave breaking (IDEMIX, Olbers and Eden, 2013).

- Support for writing output in the widely used NetCDF4 binary format (Rew and Davis, 1990), and writing restart data to pick up from a previous integration.

Veros, like pyOM2, aims to support a wide range of problem sizes and architectures. It is meant to be usable on anything between a personal laptop and a computing cluster, which calls for a flexible design, and which makes a dynamical programming language like Python a great fit for this task. Unlike pyOM2, which explicitly decomposes and distributes the model domain across multiple processes via MPI (Message Passing Interface, e.g., Gropp et al., 1999), Veros is not parallelized directly. Instead, all hardware-level optimizations are deferred to a numerical backend; currently either NumPy (Walt et al., 2011), or
Bohrium (Kristensen et al., 2013). While NumPy is commonly used, easy to install, and highly compatible, Bohrium provides a powerful runtime environment that handles high-performance array operations on parallel architectures.

The following section describes which procedures we used when translating pyOM2's Fortran code to a first, naïve Python implementation. Sect. 2.2 then outlines the necessary steps to obtain a well-performing and idiomatic, vectorized NumPy implementation. Sect. 2.3 gives an overview of some additional features that we implemented in Veros, and Sect. 2.4 finally
gives an introduction to the internal workings of Bohrium.

## 2.1    From Fortran to naïve Python

Array operations implemented in Fortran can be translated to Python / NumPy with relative ease, as long as a couple of pitfalls are avoided (such as 0-based indexing in Python vs. arbitrary indexing in Fortran). As an example, consider the following Fortran code from pyOM2:

`do j=js_pe,je_pe`





```
  do i=is_pe-1,ie_pe
    flux_east(i,j,:) = &
      0.25*(u(i,j,:,tau)+u(i+1,j,:,tau)) &
          *(utr(i+1,j,:)+utr(i,j,:))
  enddo
enddo
```

where `is_pe`, `js_pe`, `ie_pe`, `je_pe` denote the start and end indices of the current process. Translating this snippet verbatim to Python 2.7, the resulting code looks very similar:

```
for j in xrange(js_pe,je_pe):
  for i in xrange(is_pe-1,ie_pe):
    flux_east[i,j,:] = \
      0.25*(u[i,j,:,tau]+u[i+1,j,:,tau]) \
          *(utr[i+1,j,:]+utr[i,j,:])
```

In fact, we transformed large parts of the Fortran code base into valid Python by replacing all built-in Fortran constructs (such as `if`-statements and `do`-loops) by the corresponding Python syntax. We automated much of the initial translation process through simple tools like regular expressions to pre-parse the Fortran code base – e.g., the regular expression

```
do (\w)=((\w|[\+\-])+,(\w|[\+\-])+)
```

would find all Fortran `do` loops, while the expression

```
for \1 in xrange(\2):
```

replaces them with the equivalent `for` loops in Python[2]. This semi-automatic preprocessing allowed for a first working Python implementation of the pyOM2 code base after only a few weeks of coding that could be used as a basis to iterate towards a more performant (and idiomatic) implementation.

## 2.2 Vectorization

After obtaining a first working translation of the pyOM2 code, we refactored and optimized all routines for performance and readability, while ensuring consistency through continuously monitoring results. This mostly involves using vector operations instead of explicit Fortran-style loops over indices (that typically carry a substantial overhead in high-level programming languages). Since most of the operations in a finite-difference discretization consist of basic array arithmetic, a large fraction of the core routines were trivial to vectorize, such as the above example, which becomes:

```
flux_east[1:-2,2:-2,:] = \
```

---

[2]E.g. through the GNU command line tool `sed`, which is readily available on most Linux distributions.



```
0.25*(u[1:-2,2:-2,:,tau]+u[2:-1,2:-2,:,tau]) \
    *(utr[1:-2,2:-2,:]+utr[1:-2,2:-2,:])
```

Note that we replaced all explicit indices $(i,j)$ by basic slices (index ranges). The first and last two elements of the horizontal dimensions are ghost cells, which makes it possible to shift arrays by up to two cells in each dimension without

introducing additional padding. Since all parallelism is handled in the backend, there is no need to retain the special indices `is_pe`, `js_pe`, `ie_pe`, `je_pe`, and we replaced them by hard-coded values ($2$, $2$, $-2$, and $-2$, respectively).

Apart from those trivially vectorizable loops, there were several cases that required special treatment:

- Boolean masks are either cast to floating point arrays and multiplied to the to-be-masked array, or applied using NumPy's `where` function. We decided to avoid "fancy indexing" due to poor parallel performance.

- Operations where e.g., a three-dimensional array is to be multiplied with a two-dimensional array slice-by-slice can be written concisely thanks to NumPy's powerful array broadcasting functionalities (e.g., by using `newaxis` as an index).

- We vectorized loops representing (cumulative) sums or products using NumPy's `sum` (`cumsum`) and `prod` (`cumprod`) functions, respectively.

- Oftentimes, recursive loops can be reformulated analytically into a form that can be vectorized. A simple example is

$$x_{n+1}^t = 2x_n^u - x_n^t \tag{1}$$

which arises when calculating the positions $x^t$ of the T-grid cells, and that is equivalent to

$$x_{n+1}^t = (-1)^n \left( x_0^t + \sum_{i=0}^{n} (-1)^i 2x_i^u \right) \tag{2}$$

which can easily be expressed through a cumulative sum operation (`cumsum`).

On top of this, there were two loops in the entire pyOM2 codebase that were only partially vectorizable using NumPy's

current toolset[3] (such that an explicit loop over one axis remains). Since they did not have a measurable impact on overall performance, they were left in this semi-vectorized form – however, it is certainly possible that those loops (or similar future code) could become a performance bottleneck on certain architectures. In this case, an extension system could be added to Veros, where such instructions are implemented using a low-level API and compiled upon installing Veros. Conveniently, Bohrium offers zero-copy interoperability for this use-case via Cython (Behnel et al., 2011) on CPUs, and PyOpenCL and

PyCUDA (Klöckner et al., 2012) on GPUs.

---

[3]One that arises when calculating mixing lengths as in Gaspar et al. (1990) that involves updating values dynamically based on the value of the previous cell, and one inside the overturning diagnostic where a vectorization would require to temporarily store $2N_x N_y N_z^2$ elements in memory (where $N_x, N_y, N_z$ denote the number of grid elements in $x, y, z$ direction, respectively).



## 2.3 Further Modifications

Since there is an active community of researchers developing Python packages, many sophisticated tools are just one `import` statement away, and the dynamic nature of Python allows for some elegant implementations that would be infeasible or outright impossible in Fortran 90. Moving the entire code base to Python thus allowed us to implement a number of modifications that

comply with modern best practices without too much effort, some of which are described in the upcoming sections.

### 2.3.1 Dynamic Backend Handling

Through a simple function decorator, a pointer to the backend currently used for computations is automatically injected as a variable `np` into each numerical routine. This allows for using the same code for every backend, provided their interface is compatible to NumPy's. Currently, the only included backends are NumPy and Bohrium, but in principle, one could build their

own NumPy-compatible backend, e.g., by replacing some critical functions with a better performing implementation.

Since Veros is largely agnostic of the backend that is being used for vector operations, Veros code is especially easy to write – everything concerning e.g., the parallelization of array operations is handled by the backend, so developers can focus on writing clear, readable code.

### 2.3.2 Generic Stream Function Solvers

The two-dimensional barotropic stream function $\Psi$ of the vertically integrated flow is calculated in every iteration of the solver to account for effects of the surface pressure. It can be obtained by solving a two-dimensional Poisson equation of the form

$$\Delta\Psi = \int_0^{h(x,y)} \zeta(x,y,z)\,\mathrm{d}z \tag{3}$$

with coordinates $x,y,z$, total water depth $h$, vorticity $\zeta$, and Laplacian $\Delta$. The discrete version of this Laplacian in pseudo-spherical coordinates as solved by pyOM2 and Veros reads (Eden, 2014):

$$\Delta\Psi_{i,j} = \frac{\Psi_{i+1,j} - \Psi_{i,j}}{h_{i+1,j}^v \cos^2(y_j^u)\Delta x_{i+1}^t \Delta x_i^u}$$

$$- \frac{\Psi_{i,j} - \Psi_{i-1,j}}{h_{i,j}^v \cos^2(y_j^u)\Delta x_i^t \Delta x_i^u}$$

$$+ \frac{\cos(y_{j+1}^t)}{\cos(y_j^u)} \frac{\Psi_{i,j+1} - \Psi_{i,j}}{h_{i,j+1}^u \Delta y_{j+1}^t \Delta y_j^u}$$

$$+ \frac{\cos(y_j^t)}{\cos(y_j^u)} \frac{\Psi_{i,j} - \Psi_{i,j-1}}{h_{i,j}^u \Delta y_j^t \Delta y_j^u} \tag{4}$$

with

– the discrete stream function $\Psi_{i,j}$ at the $\zeta$-cell with indices $(i,j)$,

– latitude $x$ and longitude $y$, each defined at T-cells $(x_{i,j}^t, y_{i,j}^t)$ and U / V-cells $(x_{i,j}^u, y_{i,j}^u)$

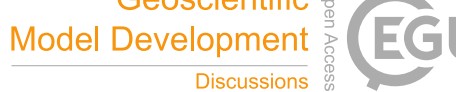



— grid spacings of T ($\Delta x_{i,j}^t$, $\Delta y_{i,j}^t$) and U / V cells ($\Delta x_{i,j}^u$, $\Delta y_{i,j}^u$) in each horizontal direction.

By re-ordering all discrete quantities $x_{i,j}$ to a one-dimensional object $x_{i+Nj}$ (with $i \in [1, N]$, $j \in [1, M]$, and $N, M \in \mathbb{N}$) and writing them as column vectors $\boldsymbol{x}$, (3) results in the equation

$$\mathbf{A}\boldsymbol{\Psi} = \boldsymbol{Z} \tag{5}$$

where $\boldsymbol{Z}$ represents the right-hand side of (3), and $\mathbf{A}$ is a banded matrix with nonzero values on five or seven diagonals[4] that reduces to the classical discrete Poisson problem for equidistant Cartesian coordinates, but is generally non-symmetric.

In pyOM2, the system (5) is solved through a conjugate gradient solver with Jacobi preconditioner in a matrix-free formulation taken from the Modular Ocean Model (MOM, Pacanowski et al., 1991). Since both the matrix-free formulation and the fixed preconditioner lead to a quite specific solver routine, our first step was to transform this into a generic problem by incorporating all boundary conditions into the actual Poisson matrix, and to use `scipy.sparse` from the SciPy library (Jones et al., 2001–) to store the resulting banded matrix. At this stage, any sufficiently powerful sparse linear algebra library can be used to solve the system. This is especially important for Veros as it is targeting a wide range of architectures – a small, idealized model running with NumPy does not require a sophisticated algorithm (and can stick with e.g., the readily available solvers provided by `scipy.sparse.linalg`); intermediate problem sizes might require a strong, sequential algorithm; and for large setups, highly parallel solvers from a high-performance library are usually most adequate (such as PETSc (Balay et al., 1997) on CPUs, or CUSP (Dalton et al., 2014) on GPUs).

In fact, as shown in Sect. 3.2, substantial speedups could be achieved by using an AMG[5] preconditioner provided by the Python package pyAMG (Bell et al., 2013). Even though the AMG algorithms are mathematically highly sophisticated, pyAMG is simple to install (e.g., via the PyPI package manager, `pip`), and implementing the preconditioner into Veros required merely a few lines of code, making this process a prime example for the huge benefits one can expect from developing in a programming language as popular in the scientific community as Python. And thanks to the modular structure of the new Poisson solver routines, it will be easy to switch (possibly dynamically) to even more powerful libraries as it becomes necessary.

### 2.3.3   Multi-threaded I/O with Compression

In geophysical models, writing model output or restart data to disk often comes with its own challenges. When output is written frequently, significant amounts of CPU time may be wasted waiting for disk operations to finish. Additionally, data sets tend to grow massive in terms of file size, usually ranging from Gigabytes to Petabytes. To address this, we took the following measures in Veros:

— All disk output is written in a separate thread, enabling computations to continue without waiting for flushes to disk to finish.

---

[4]Two additional diagonals are introduced when using cyclic boundary conditions to enforce $\Psi_{N,j} = \Psi_{0,j}$ $\forall j \in [0, M]$

[5]Algebraic Multi-Grid (Vaněk et al., 1996)


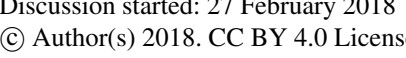


– By default, Veros makes use of NetCDF4 and HDF5's built-in compression abilities. Simply by passing the desired compression level as a flag to the respective library, the resulting file sizes were reduced by about two thirds, with little computational overhead. Since the zlib (NetCDF4) and gzip (HDF5) compression is built into the respective format specification, most standard post-processing tools are able to read and decompress data on the fly, without any explicit
user interaction.

### 2.3.4 Backend-specific Tridiagonal Matrix Solvers

Many dissipation schemes contain implicit contribution terms, which usually requires the solution of some linear system $A\boldsymbol{x} = \boldsymbol{b}$ with a tridiagonal matrix $A$ for every horizontal grid point (e.g., Gaspar et al., 1990; Olbers and Eden, 2013).

In pyOM2, those systems are solved using a naïve Thomas algorithm (simplified Gaussian elimination for tridiagonal sys-
tems). This algorithm cannot be fully vectorized with NumPy's toolkit, and explicit iteration turned out to be a major bottleneck for simulations. One possible solution was to re-write all tridiagonal systems for each horizontal grid cell into one large, padded tridiagonal system that could be solved in a single pass. This proved to be feasible for NumPy, since it exposes bindings to LAPACK's `dgtsv` solver (Anderson et al., 1999), but performance was not sufficient when using Bohrium. We therefore made use of Bohrium's interoperability functionalities, which allowed us to implement the Thomas algorithm directly in the OpenCL
language for high-performance computing on GPUs via PyOpenCL (Klöckner et al., 2012); on CPUs, Bohrium provides a parallelized C-implementation of the Thomas algorithm as an "extension method".

When encountering such a tridiagonal system, Veros automatically chooses the best available algorithm for the current runtime system (backend and hardware target) without manual user interaction. This way, overall performance increased substantially, to the levels reported in Sect. 3.2.

### 2.3.5 Modular Diagnostic Interface

All model diagnostics, such as snapshot output, vertical (overturning) stream functions, energy flux tracking, and temporal mean output, are implemented as subclasses of a diagnostics base class, and instances of these subclasses are added to a Veros instance dynamically. This makes it possible to add, remove, and modify diagnostics on the fly:

```
def set_diagnostics(self):
    diag = veros.diagnostics.Average()
    diag.name = "annual-mean"
    diag.output_frequency = 360 * 86400
    self.diagnostics["annual-mean"] = diag
```

This code creates a new averaging diagnostic that outputs annual means, and can be repeated, e.g., for also writing monthly
means.



Besides enforcing a common interface, creating all diagnostics as subclass of a "virtual" base class also has the benefit that common operations like data output are defined as methods of said base class, providing a complete and easy-to-use toolkit to implement additional diagnostics.

### 2.3.6 Metadata Handling

About 2000 of the approximately 11000 SLOC (source lines of code) in pyOM2 were dedicated to specifying variable metadata (often multiple times) for each output variable, leaving little flexibility to add additional variables, and risking inconsistencies. In Veros, all variable metadata is contained in a single, central dictionary; subroutines may then look up metadata from this dictionary on demand (e.g., when allocating arrays, or preparing output for a diagnostic). Additionally, a "cheat sheet" containing a description of all model variables is compiled automatically and added to the online user manual.

This approach maximizes maintainability through eliminating inconsistencies, and allows users to add custom variables that are treated no differently from the ones already built-in.

### 2.3.7 Quality Assurance

To ensure consistency with pyOM2, we developed a testing suite that runs automatically for each commit to the master branch of the Veros repository. The testing suite is comprised of both unit tests and system tests:

**Unit tests** are implemented for each numerical core routine; they call a single routine with random data and make sure that results match between Veros and pyOM2 within a certain tolerance.

**System tests** integrate entire model setups for a small number of time steps and compare the results to pyOM2.

These automated tests allow developers to detect breaking changes early and ensure consistency for all numerical routines and core features apart from deliberately breaking changes. To achieve strict compliance with pyOM2 during testing, we 20 introduced a compatibility mode to Veros that forces all subroutines to comply with their pyOM2 counterpart, even if the original implementation contains errors that we corrected when porting them to Veros.

  Using this compatibility mode, the results of most of Veros' core routines match those of PyOM2 within a global, absolute tolerance of $10^{-8}$, while in a few cases an accuracy of just $10^{-7}$ is achieved (presumably due to a higher sensitivity to round-off errors of certain products). The longer-running system tests achieve global accuracies between $10^{-6}$ and $10^{-4}$ for all model 25 variables. All results are normalized to unit scale by dividing by their global maximum before comparing.

### 2.4 About Bohrium

Since Veros relies heavily on the capabilities of Bohrium for large problems on parallel architectures, this section gives a short introduction to the ideas behind and implementation of Bohrium.

  Bohrium is a software framework for efficiently mapping array-operations from a range of frontend languages (currently 30 C, C++, and Python) to various hardware architectures, including multi-core CPUs and GPGPUs (Kristensen et al., 2013).

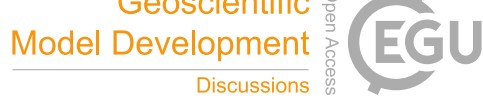



The components of Bohrium are outlined in Larsen et al. (2016): All array operations called by the frontend programming languages are passed to the respective bridge, which translates all instructions into Bohrium bytecode. After applying several bytecode optimizations, it is compiled into numerical kernels which are then executed a the backend. Parallelization is handled by so-called vector engines, currently using OpenMP (Dagum and Menon, 1998) on CPUs, and either OpenCL (Stone et al.,

2010) or CUDA (Nickolls et al., 2008) on GPUs.

Since Bohrium uses lazy evaluation, successive operations on the same array views can be optimized substantially. On one hand, operations can be reordered or simplified analytically to reduce total operation counts. On the other hand, a sophisticated *fusion algorithm* is applied, which "is a program transformation that combines (fuses) multiple array operations into a kernel of operations. When it is applicable, the technique can drastically improve cache utilization through temporal data locality and

enables other program transformations, such as streaming and array contraction (Gao et al., 1993)" (Larsen et al., 2016). In fact, this fusion algorithm alone may increase performance significantly in many applications (Kristensen et al., 2016).

Bohrium's Python bridge is designed to be a drop-in replacement for NumPy, supplying a multi-array class `bohrium.ndarray` that derives from NumPy's `numpy.ndarray`. All array meta-data is handled by the original NumPy, and only actual computations are passed to Bohrium, e.g., when calling one of NumPy's "ufuncs" (universal functions). This way, most of NumPy's

functionality is readily available in Bohrium[6], which allows developers to use Bohrium as a high-performance numerical backend while writing hardware-agnostic code (and leaving all optimizations to Bohrium). These properties make Bohrium an ideal fit for Veros.

## 3  Verification & Performance

### 3.1  Consistency Check

Since all Veros core routines are direct translations of their pyOM2 counterparts, an obvious consistency check is to compare the output of both models. On a small scale, this is already done in the Veros testing suite, which ensures consistency for most numerical routines in isolation, and for a few time steps of the model as a whole (see Sect. 2.3). However, real-world simulations often run for anything between thousands and millions of iterations, possibly allowing numerical roundoff or minor coding errors to accumulate to significant deviations.

In order to check whether this is a concern in our case, we integrated a global model setup with coarse resolution (approx. $4° \times 4°$, $90 \times 40 \times 15$ grid elements) for a total of 50 model years (18000 iterations) using Veros with NumPy, Veros with Bohrium, and pyOM2. The resulting output reveals no physically significant deviation between either of the simulations, with maximum relative errors of about $10^{-4}$ (Veros, between NumPy and Bohrium) and $10^{-6}$ (between Veros with NumPy and pyOM2 when using the compatibility mode).

---

[6]Except NumPy functions implemented in C, which have to be re-implemented inside Bohrium to be available.





|  | Desktop PC (I) | Cluster Node (II) |
|---|---|---|
| CPU | Intel® Core™ i7 6700 @ 3.40 GHz (4 physical / 8 logical cores) | 2 × Intel® Xeon® E5-2650 v4 @ 2.20 GHz (24 physical / 48 logical cores) |
| RAM | 16 GB DDR4 | 512 GB DDR4 |
| Storage | M.2 SSD @ 500 MBs$^{-1}$ read/write performance | LUSTRE filesystem @ 128 MBs$^{-1}$ read/write performance |
| GPU | — | Nvidia Tesla P100 (16 GB HBM2 memory) |
| Software stack | GNU compiler toolchain 7.2.0, Python 2.7, NumPy 1.13.3, Bohrium 8.9.0 | GNU compiler toolchain 5.4.0, CUDA 9.0, Python 2.7, NumPy 1.13.3, Bohrium 8.9.0 |

**Table 1.** Specifications of the two benchmark architectures.

## 3.2 Benchmarks

As high-performance computing resources are still expensive and slow model execution is detrimental to a researcher's work-flow, performance is of course a critical measure for any geophysical model (and usually the biggest counter-argument against using high-level programming languages in modelling). It is thus essential to try and measure the performance of Veros through
benchmarking, and since we are in the lucky position to have a well-performing reference implementation available, an obvious test is to compare Veros' throughput to pyOM2's.

To this end, we developed a benchmarking suite that is part of the Veros code repository, so that benchmarks can easily be executed and verified on various architectures. These benchmarks consist of either complete model runs or single subroutines that are executed with varying problem sizes for each of the available numerical backends (NumPy, Bohrium, and pyOM2's
Fortran library with and without MPI support)[7].

The benchmarks were executed on two different architectures: a typical Desktop PC, and a cluster node, marked as architecture I and II, respectively (Table 1). Note that, since Bohrium does not yet support distributed memory architectures, comparisons have to stay confined to a single computational node. Bohrium v0.8.9 was compiled from source with GCC and `BUILD_TYPE=Release` flags, and pyOM2 with gfortran using `-O3` optimization flags and OpenMPI support.

### 3.2.1 Overall Performance

In order to benchmark the overall performance of Veros against that of pyOM2, an idealized model setup consisting of an enclosed basin representing the North Atlantic with a zonal channel in the south is integrated for a fixed number of 100 iterations, but with varying problem sizes, for each numerical backend.

---

[7]Since pyOM2 offers Python bindings through f2py for all of its core routines, it can actually be used as a Veros backend. This way, we can ensure that all components solve the exact same problem.



The results (Fig. 1) show that:

- For large problems with a number of total elements exceeding $10^7$ (which is about the number of elements in a global setup with $1° \times 1°$ horizontal resolution), the Bohrium backend is at its peak efficiency and about 2.3 times slower than parallel pyOM2, regardless of the number of CPU cores. Running on architecture II's high-end GPU, Veros' throughput is comparable to that of pyOM2 running on 24 CPUs.

- Veros' NumPy backend is about 3 times slower than pyOM2 running serially, largely independent of the problem size.

- For small problems containing $\lesssim 2 \times 10^4$ elements, parallelism is inefficient, so NumPy performs relatively well.

- Using Bohrium carries a high overhead, and it only surpasses NumPy in terms of speed for problems larger than about $10^5$ elements.

- Veros is least efficient for intermediate problem sizes of about $10^5$ elements (up to 50 times slower than parallel pyOM2 on 24 CPUs).

We believe that these performance metrics show that Veros is indeed usable as the versatile ocean simulator it is trying to be. Even students without much HPC experience can use Veros to run small to intermediate-sized, idealized models through NumPy, and seamlessly switch to Bohrium later on to run realistic, full-size setups while experiencing performance comparable to traditional ocean models. And given that Bohrium is still undergoing heavy development, we expect that many of the current limitations will be alleviated in future versions, causing Veros to perform even better than today.

### 3.2.2 Stream Function Solver

To illustrate the speedups that could be achieved for the stream function solver alone (Sect. 2.3), we conducted similar benchmarks calling only the corresponding solvers in pyOM2 and Veros using pseudo-spherical coordinates, uniform grid spacings, cyclic boundary conditions, and a solver tolerance of $10^{-12}$, for a total of 100 times with different, random right-hand-side vectors.

The results show that Veros' stream function solver easily beats pyOM2's for most relevant problem sizes (Fig. 2), even though the underlying BiCGstab solver `scipy.sparse.linalg.bicgstab` is not parallelized (apart from internal calls to the multi-threaded OpenBLAS library for matrix-vector products). The credit for this speedup belongs entirely to pyAMG, as the AMG preconditioner causes much faster convergence of the iterative solver.

When running on an even higher number (possibly hundreds) of CPU cores, pyOM2's parallel conjugate gradient solver can be expected to eventually outperform Veros' serial AMG solver. However, thanks to the new, generalized structure of the stream function routines (Sect. 2.3), the SciPy BiCGstab solver could easily be switched with a different, parallel library implementation.



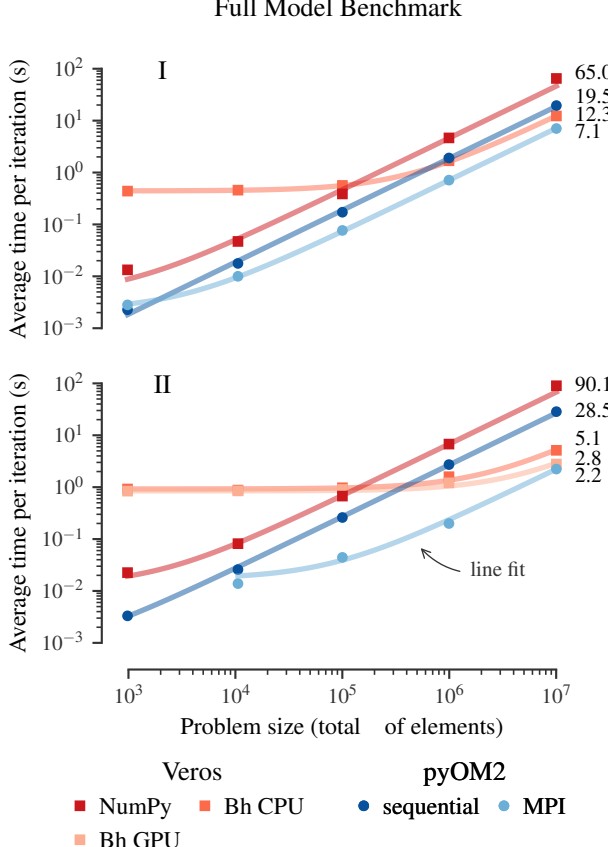

**Figure 1.** In terms of overall performance, Veros using Bohrium (Bh) is slower than pyOM by a factor of about 1.3 to 2.3 for large problems, depending on the hardware architecture (I and II, cf. Table 1). Line fits suggest a linear scaling with constant overheads for all components.

## 4 Application: Kelvin Wave Propagation

In the current literature we see a gap between theory and very idealized models on one hand, and primitive equation models with realistic forcing and topography on the other hand. Here, we will apply Veros to an aspect of the Southern Ocean (SO) hypothesis by Toggweiler and Samuels (1995).

They propose that a strengthening of SO winds leads to a strengthening of the Atlantic Meridional Overturning Circulation (AMOC). Their main argument is based on geostrophy and mass conservation, and it states that mass pushed north by the Atlantic Ocean Ekman layer has to be replaced by upwelled water from depths below the Drake Passage sill. This basic idea is largely accepted, and much of the discussion in the literature is now quantitative, i.e., how much of the wind-driven Eulerian transport in the SO is compensated by meso-scale eddy-driven transport of opposite sign (Munday et al., 2013). However, Jochum and Eden (2015) show that in at least one general circulation model (GCM) the AMOC does *not* respond to changes in





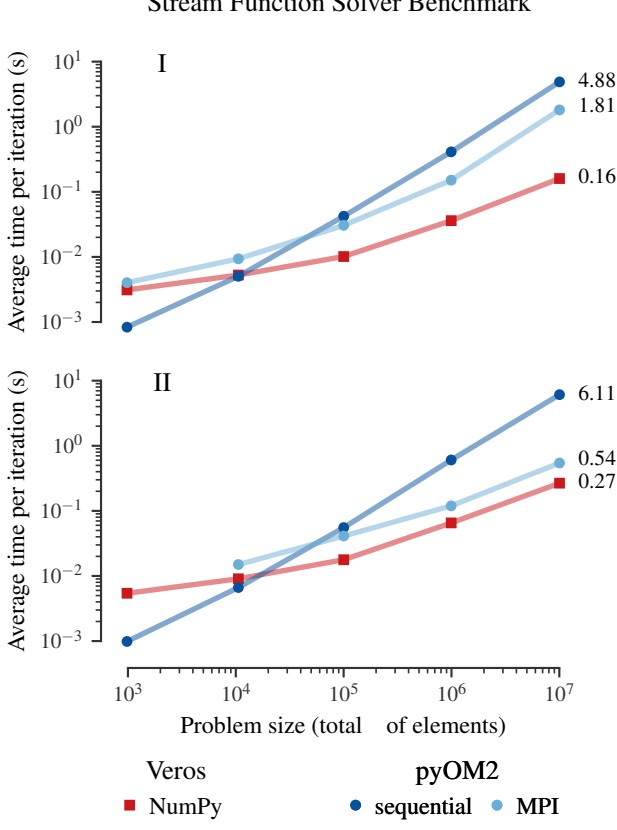

**Figure 2.** Thanks to pyAMG's AMG preconditioner, Veros' stream function solver is between 2 (24 CPUs, II) and 11 (4 CPUs, I) times faster than pyOM2's parallel conjugate gradient solver for large problem sizes.

SO winds. Thus, testing the Southern Ocean hypothesis requires us not only to test if ocean models represent mesoscale eddies appropriately, but also if the propagation of SO anomalies into the northern hemisphere is simulated realistically.

The main propagation mechanism is planetary waves; changes to SO Ekman divergence and convergence set up buoyancy anomalies that are radiated as Kelvin and Rossby waves and set up changes to the global abyssal circulation (McDermott,

5   1996). Because they are so important there is a large literature devoted to the fidelity of planetary waves in ocean models: For example, Hsieh et al. (1983) and Huang et al. (2000) show that even coarse resolution ocean models can support meridionally propagating waves similar to Kelvin waves, and Marshall and Johnson (2013) quantify how exactly numerical details will affect wave propagation. We wish to bridge the gap between these idealized studies and GCMs by investigating the dependence of Kelvin wave phase speed on resolution in Veros. While this is in principle a minor exercise suitable for undergraduate

10  students, the presence of internal variability and irregular coastlines makes this a major challenge (Getzlaff et al., 2005). Veros considerably simplifies this problem by providing uncomplicated ways to modify the coastline, as outlined in the upcoming section. Accordingly, we use this setup for 3 month BSc projects.





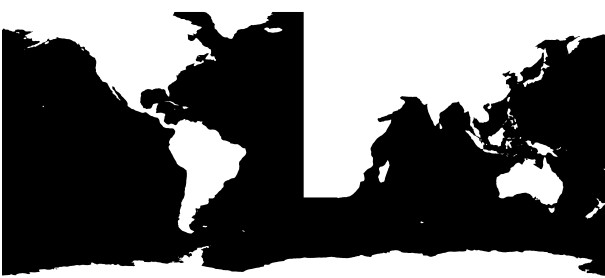

**Figure 3.** Idealized, binary geometry mask for the Kelvin wave study.

## 4.1 Modified Geometry with flexible Resolution

Modifying the geometry of a realistic geophysical model is no trivial task, especially when allowing for a flexible number of grid elements. Any solution that converts cells from water to land or vice-versa has to infer reasonable values for initial conditions and external forcing at these cells, since, e.g., atmospheric conditions tend to differ fundamentally between water
and land.

To automate this process, we created a downsampled version of the ETOPO1 global relief model (Amante and Eakins, 2009), which we exported as a binary mask indicating either water or land. We then manually edited this mask using common image processing software by removing lakes and inland seas, thickening Central America, and converting the eastern boundary of the Atlantic to a straight meridional line, running from the southern tip of Africa to the Arctic (Fig. 3).
This binary mask is read by Veros during model setup, and interpolated to the chosen grid (number of grid cells in each dimension are defined by the user; grid steps are chosen to minimize discretization error according to Vinokur (1983)). The ocean bathymetry is read from the same downsampled version of ETOPO1, and cells are converted between water and land according to the interpolated mask.

Since all cells that were converted from land to water lie in the North Atlantic, it is sufficient to modify initial conditions
and atmospheric forcing in this region only. Initial conditions are read from a reference file with $1° \times 1°$ horizontal resolution, and interpolated bilinearly to the modified grid. The bathymetry in the Atlantic is replaced by a constant depth of $4000\,\mathrm{m}$. Optionally, a different constant depth and/or linear slope for some distance from each coast can be added to model a continental shelf. All atmospheric forcing is replaced by its zonal mean value in the Atlantic basin.

This leaves us with a modified setup that is smooth enough to be stably integrated, and that allows us to track Kelvin waves
in a more isolated environment. As a first sanity check, the resulting ocean circulation looks largely as expected (Fig. 4).

## 4.2 The Experiment

If coarse resolution ocean models can support Kelvin-wave-like features, the question of phase speed becomes paramount: A wave that is too slow will be damped away too early and inhibit oceanic teleconnections, which may cause different observed



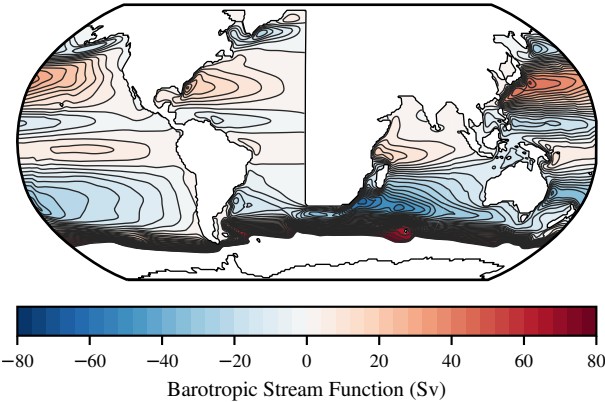

**Figure 4.** Long-term average barotropic stream function (BSF) of a $1° \times 1°$ horizontal resolution setup as described in Sect. 4.1. $1\,\mathrm{Sv} = 10^6\,\mathrm{m^3s^{-1}}$. Contours are drawn in steps of $4\,\mathrm{Sv}$.

climate sensitivities in different climate models (Greatbatch and Lu, 2003). Hsieh et al. (1983) discuss in great detail how choices in the numerical setup modify the phase speed of Kelvin waves: resolution, friction, discretization (Arakawa B or C grid, Arakawa and Lamb, 1977) and boundary conditions all affect the phase speed. However, Marshall and Johnson (2013) point out that for an adjustment timescale on the order of years or longer (relevant for Toggweiler and Samuels' SO hypothesis),

the corresponding waves have the properties of Rossby waves, albeit with a phase speed of $c = L_d/\delta_M$, where $c$ is the Kelvin wave phase speed, $L_d$ is the Rossby radius of deformation and $\delta_M = (\nu/\beta)^{1/3}$ the Munk boundary layer width. Here we test their analytical result, particularly whether the phase speed really depends only on friction but not resolution.

The global setup of Veros is used in two configurations: $2°$ (2DEG) and $1°$ zonal resolution. Both have 180 meridional grid cells with a spacing of approximately $0.5°$ at the equator and $1.5°$ at the poles. The $1°$ setup is used with two different

viscosities: $5 \times 10^4\,\mathrm{m^2s^{-1}}$ (same as 2DEG) and $5 \times 10^3\,\mathrm{m^2s^{-1}}$, called 1DEG and 1DEGL, respectively. Each of these three setups is initialized with data from Levitus (1994) and integrated for 60 years (these are our 3 control integrations). After 50 years, one new integration is branched off from each, with the maximum winds over the SO increased by $50\%$ (sine envelope between $27°$S and $69°$S). The velocity fields are sampled as daily means, and by analyzing at $200\,\mathrm{m}$ depth the differences of the first 150 days to the first 150 days of year 51 of the control integrations, we arrive at an estimate of the speed with which

the information of the SO wind stress anomaly travels north along the eastern boundary of the Atlantic Ocean.

As a first step we confirm that the anomaly signal is well resolved along the equator. Indeed, for all three setups we find the same phase speed of $2.7\,\mathrm{ms^{-1}}$ (Fig. 5a, only 1DEG is shown), slightly less than the $2.8\,\mathrm{ms^{-1}}$ that is expected from theory and observations (Chelton and coauthors, 1998). Along the African coast we find a similar speed in 1DEGL, but slower in 1DEG and 2DEG (Fig. 5b-d). Using the approximate slope of the propagating signal's contours as a metric for the average phase

speed between the equator and $40°$N, we arrive at about $2.1\,\mathrm{ms^{-1}}$ for 1DEGL, and $1.0\,\mathrm{ms^{-1}}$ for 2DEG and 1DEG.

The Rossby radius of deformation $L_d$ along the western coast of North Africa changes from approximately $100\,\mathrm{km}$ at $5°$N to $40\,\mathrm{km}$ at $30°$N (Chelton and coauthors, 1998). The Munk boundary layer width $\delta_M$ for our two different viscosities are



130 km and 60 km. The large range of $L_d$ along the coast makes it difficult to determine the exact theoretically expected phase speed, but based on Marshall and Johnson (2013) one can expect that the anomalies generated by a SO wind perturbation travel slower by a factor of less than 3 in 2DEG *and* 1DEG, and twice as fast than that in 1DEGL. This is exactly what is found here.

This minor initial application demonstrates how Veros can be used to bridge the gap between theory and full ocean GCMs. Future studies will investigate in more detail the interaction between the anomalies traveling along the coast and high latitude stratification and topography.

## 5 Summary & Outlook

By translating pyOM2's core routines from Fortran 90 to vectorized Python / NumPy code (Sect. 2.1 and Sect. 2.2), and adding integration with the Bohrium framework (Sect. 2.4), we were able to build a Python ocean model (Veros) that is both consistent with pyOM2 to a high degree (Sect. 3.1), and does not perform significantly worse, even on highly parallel architectures (Sect. 3.2). Additional modifications (Sect. 2.3) include a powerful algebraic multigrid (AMG) Poisson solver, compressed NetCDF4 output, a modular interface for diagnostics, self-documentation, and automated testing.

A simple experiment investigating planetary wave propagation in the Atlantic showed that boundary waves in GCMs travel with phase speeds consistent with theoretical expectations.

While creating Veros did require a deep understanding of the workings of NumPy and Bohrium to avoid performance bottlenecks and to write concise, idiomatic, vectorized code, the presented version of Veros took less than a year to develop by a single, full-time researcher. Nevertheless, Veros is still at an early stage of development. In future releases, we plan to address the following issues:

**More abstraction** Most of Veros' core routines are currently direct vectorized translations of pyOM2's Fortran code, which manipulate array objects through basic arithmetic and provide little exposition of the underlying numerical concepts. In order to create a truly approachable experience, it is crucial to deviate from this approach and introduce more abstraction by grouping common patterns into higher-order operations (like transpositions between grid cell types or the calculation of gradients).

**Parallelized stream function solvers** A parallel Poisson solver is a missing key ingredient to scale Veros efficiently to even larger architectures. Solvers could either be provided through Bohrium, or by binding to another third-party library such as PETSc (Balay et al., 1997), ViennaCL (Rupp et al., 2010), or CUSP (Dalton et al., 2014).

**Distributed memory support** High-resolution representations of the ocean (such as eddy-permitting or eddy-resolving models) are infeasible to be simulated on a single machine, since the required integration times may well take decades to compute. In order for Veros to become a true all-purpose tool, it is crucial that work can be distributed across a whole computing cluster (which could either consist of CPU or GPU nodes). Therefore, providing distributed memory support either through Bohrium or another numerical backend is a top priority for ongoing development.



However, we think that Veros has proven that it is indeed possible to implement high-performance geophysical models entirely in high-level programming languages.

*Code availability.* The entire Veros source code is available under a GPL license on GitHub (https://github.com/dionhaefner/veros). All comparisons and benchmarks presented in this study are based on the Veros v0.1.0 release, which is available under the DOI 10.5281/zenodo.1133130. The model configuration used in Sect. 4 is included as a default configuration ("wave propagation").

The Veros user manual is hosted on ReadTheDocs (https://veros.readthedocs.io). An archived version of the Veros v0.1.0 manual, along with the user manual of PyOM2 describing the numerics behind Veros, is found under the DOI 10.5281/zenodo.1174390.

Recent versions of PyOM2 are available at https://wiki.cen.uni-hamburg.de/ifm/TO/pyOM2. A snapshot of the pyOM2 version Veros is based on, and that is used in this study, can be found in the Veros repository.

*Competing interests.* The authors declare that they have no conflict of interest.

*Acknowledgements.* DH was supported through PREFACE (EU FP7/2007-2013), under grant agreement no. 603521; all others were supported through the Niels Bohr Institute. Computational resources were provided by $DC^3$, the Danish Center for Climate Computing. We kindly thank everyone involved in the development of Veros, and the scientific Python community for providing the necessary tools.

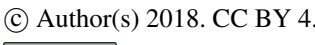




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





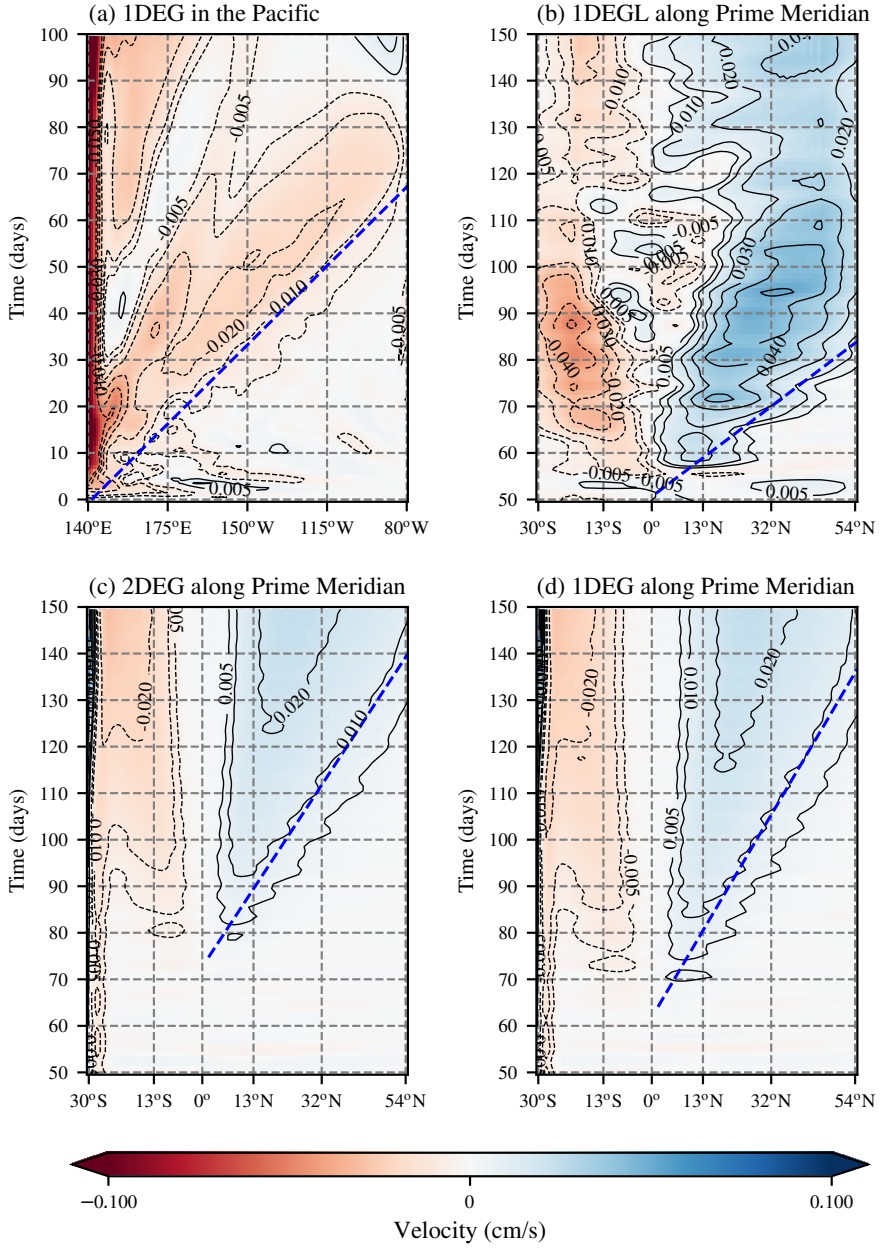

**Figure 5.** The Kelvin wave phase speed in Veros approximately only depends on viscosity, not resolution, as predicted by Marshall and Johnson (2013). Shown is (a) zonal velocity in 1DEG at the Equator, and (b–d) meridional velocity in 1DEGL, 2DEG, 1DEG, respectively, along 0° longitude in the Atlantic at 200 m depth. The slopes of the blue dashed lines are used to estimate phase speeds. Note that the signal arrives at different times at the African coast due to the location of the maximum wind field perturbation, which is not at the South American coast. Thus, the buoyancy perturbation that eventually arrives in the North Atlantic has to be advected to the South American coast before it can travel north as a fast coastal wave.