# Peer review of "Veros v0.1 — a Fast and Versatile Ocean Simulator in Pure Python"

_Geoscientific Model Development, 2018_

## Referee Comment (RC1) · Anonymous Referee #1 · 10 Apr 2018

**General comments**

The manuscript is introducing a new generation of ocean circulation model coded in Python. If it is the first Python code for those applications, it would be intersting to mention it. If not, it would be relevant to introduce other similar applications.

Veros is based on a wide range of Python libraries. As this code will be used, for example, for educational purpose, can the authors detail the code management plan considering potential near future compatibility issues between those libraries ?

The Veros code is developed for single-node computation. Can the authors discuss the potential near future extent of the code for parallel computing (several nodes) and then more expensive applications ?

Despite the description of advantages of Python language, it seems that the code is mainly designed for educationnal purpose. Could the authors confirm this or detail those advantages in the manuscript ?

A last general comment is referring to the experiment. The choice of the model grid is surprising. It is not self explanatory why the straight meridional line is used in the Atlantic. It seems that coastline modification is manually modified outside Veros code and, then, it does not show Veros extended functionnality as mentionned in section 4 (p. 14 « uncomplicated ways to modify the coastline »).

**Specific comments**

Page 1 :

line 5 : Please add « global » before « ocean model » as using a coarse resolution (1° x 1°), 15 vertical levels over a global ocean are needed to reach one millions points.

Page 2 :

lines 5-10 : The authors suggest that there are more possible errors in Fortran programming. However, this is related to the technical rigour of the developper/user and of the strict application of good practices and Fortran norms as it should for developping a Python code. Please consider rephrasing thoses lines.

Page 4 :

ligne 8 : Why don't use Python 3.x which is now mature and ready to use ?

Page 6 :

paragraph 2.3 : using good practice, Fortran code could also be elegant and easily readable…
Keep in mind that Fortran means **FOR**mula **TRAN**slator ! Please consider to be more factual
in your remarks

Page 7 :

line 11-16 : Would it mean that user should use the appropriate algebra library depending
on the size of the problem ? Numpy, PETSc, CUSP ? Or Veros chooses automatically the best
one like in section 2.3.4 ?

line 17 : The authors refer to a following section. Could you consider to give more details to
improve the readability ?

lines 29-30 : Indeed, I/O management is a main issue in many codes for now and in the
future. Give more details on this output strategy.

Page 8 :

Line 1-5 : Is there a loss in accuracy with compress and decompress processes ?

Page 9 :

Line 15 : « certain tolerance » : Please, could you be more explicit on this point ?

Page 10 :

Line 28 : Which variables do you consider for those relative errors ?

Page 11 :

Line 13 : It sounds that a main drawback is that Borhium an only be used in a single
computational node. Have you any idea the schedule for parallelized implementation of
Bohrium ?

Page 12 :

Figure 1.
   -   Bh CPU and Bh GPU curves can not be distinguished.
   -   I do not clearly understand what does the « line fit » for the MPI curve ?

**Technical corrections**

Page 6 :

line 12 : it seems the character – at the beginning at the line is not necessary.

Page 7 :

line 11 : it seems the character – in reference Jones et al is not necessary. (Idem page 20 line16)

line 17 : Move the AMG citation in the main text instead of in the footer note.

Page 10 :

Line 22 : « Sec. 2.3.7 » instead of « Sect. 2.3 »

Page 14 :

Line 5 : Replace « in ocean models : For » by « in ocean models. For »

---

## Author Comment (AC1) · 21 Apr 2018

Dear referee,

thank you very much for your detailed review. We shall address the issues you raised point-by-point below. A 'latexdiff' of the revised manuscript is attached as a supplement to this response.

Please let us know should you have further questions.

On behalf of the authors,

Dion Häfner

[Figure]

**1 General comments**

The manuscript is introducing a new generation of ocean circulation model coded in Python. If it is the first Python code for those applications, it would be intersting to mention it. If not, it would be relevant to introduce other similar applications.

To our knowledge, Veros is the first serious approach to a pure Python ocean model. There are several projects that provide a Pythonic front-end to some wrapped Fortran code, such as CliMT [1], OOF$\varepsilon$ [2], or Veros' parent project, PyOM [3], but none that go all the way. This may be connected to the fact that there is no obvious "right" way to parallelize Python / NumPy code. I added a paragraph to the introduction.

Veros is based on a wide range of Python libraries. As this code will be used, for example, for educational purpose, can the authors detail the code management plan considering potential near future compatibility issues between those libraries ?

In an ecosystem as volatile as the scientific Python stack, proper dependency management is indeed important. So far we have only experienced minor compatibility issues, which were easily fixed by requiring certain minimum versions of dependencies in 'setup.py'. We thus chose not to discuss this in the manuscript. In case dependency issues should become a major concern in the future, we would probably resort to a package manager like 'conda', and supply an official 'conda' recipe for Veros (so people could use 'conda install veros' to get a working copy of all dependencies and the Veros code).

The Veros code is developed for single-node computation. Can the authors

discuss the potential near future extent of the code for parallel computing (several nodes) and then more expensive applications ?

Implementing distributed memory support in Bohrium won't be a trivial task, but it could be done (see below). Apart from that, there are other libraries we could leverage for multi-node support, such as 'dask.array' in conjunction with 'dask.distributed'. However, distributed simulation in Python is not very mature, and we don't know yet how any given solution will perform in practice. We thus don't feel comfortable to speculate too much about a possible time frame.

Despite the description of advantages of Python language, it seems that the code is mainly designed for educationnal purpose. Could the authors confirm this or detail those advantages in the manuscript ?

Yes and no. Veros is built for getting rapid insights into how the ocean works. This is of course very useful for students, but there are still many fundamental open questions regarding ocean mechanics that are to be addressed on a research level, and where traditional models might be too inflexible. I added a sentence to the introduction.

A last general comment is referring to the experiment. The choice of the model grid is surprising. It is not self explanatory why the straight meridional line is used in the Atlantic.

The transition to section 4.1 was indeed a bit harsh. I have added a paragraph on our motivation.

It seems that coastline modification is manually modified outside Veros code and, then, it does not show Veros extended functionnality as mentionned in section 4 (p. 14 Âń uncomplicated ways to modify the coastline Âż)

The sentence you quoted was indeed poorly worded. I have added a paragraph. Keep in mind that we do *not* modify the input data sets for bathymetry, forcing fields, and intial conditions. All we supply is a binary mask image, and Veros does the rest (including interpolating everything to the chosen, flexible, resolution).

**2  Specific comments**

Page 1 :
line 5 :  Please add Âń global Âż before Âń ocean model Âż as using a coarse resolution (1° x 1°), 15 vertical levels over a global ocean are needed to reach one millions points.

Agreed.

Page 2 :
lines 5-10 :  The authors suggest that there are more possible errors in Fortran programming. However, this is related to the technical rigour of the developper/user and of the strict application of good practices and Fortran norms as it should for developping a Python code.  Please consider rephrasing thoses lines.

I agree that we did not convey our point very well here. It is probably true that an expert Fortran programmer lets *fewer* bugs slip into production code than an expert Python programmer, if only due to the strict type checking provided by the Fortran compiler. However, there are several things to consider:

- People implementing new features such as a novel parameterization are, in our experience, often *not* expert Fortran programmers, but grad students or postdocs more

interested in Physics than software engineering. - While it is in principle possible to write clear Fortran code with meaningful abstractions that may be just as readable as a high-level implementation, the reality is often different. Popular ocean models such as MOM [4] or POP2 [5] feature subroutines that are hundreds to thousands of lines long, and both models rely on more obscure Fortran features such as 'COMMON' blocks, which makes it hard to keep track of variable scopes for inexperienced programmers. This is not necessarily due to flaws in Fortran's core design, but we do consider the established idiomatic style of a community to be tightly bound to the language used.

I have re-worded this section a bit.

> Page 4 :
> ligne 8 : Why don't use Python 3.x which is now mature and ready to use ?

The first prototype of Veros was written in Python 2.7 to work around some bugs in Bohrium at the time. Those issues have since been resolved, and we fully support both Python 2.7 and Python 3.x. I have removed all Python 2 references from the manuscript, as it was not really relevant.

> Page 6 :
> paragraph 2.3 : using good practice, Fortran code could also be elegant and
> easily readable... Keep in mind that Fortran means FORmula TRANslator !
> Please consider to be more factual in your remarks

While Fortran might indeed be a good choice to translate formulae, this section specifically deals with the ecosystem around the numerical core of a simulation project: third-party library integration, dynamic switching between modules during run time, modern productivity and QA tools, modularity and object-oriented programming. Since Fortran 90 does not support classes, has very few users outside of academia, and is entirely

static, I think it is safe to say that Python allows for some elegant implementations that are infeasible or outright impossible in Fortran 90.

> Page 7 :
> line 11-16 : Would it mean that user should use the appropriate algebra library depending on the size of the problem ? Numpy, PETSc, CUSP ? Or Veros chooses automatically the best one like in section 2.3.4 ?

Whether to use NumPY, PETSc, or CUSP depends more on the available hardware and software architecture than the size of the problem, so it would make sense to pick the most appropriate one automatically as it is done with the tridiagonal solver. But since we don't know the performance characteristics of those libraries yet, we can only speculate at this point.

> line 17 : The authors refer to a following section. Could you consider to give more details to improve the readability ?

I made this section a bit more descriptive.

> lines 29-30 : Indeed, I/O management is a main issue in many codes for now and in the future. Give more details on this output strategy.

Done.

> Page 8 :
> Line 1-5 : Is there a loss in accuracy with compress and decompress processes ?

Both compression algorithms ('zlib' and 'gzip') are lossless.

> Page 9 :
> Line 15 : Âń certain tolerance Âż : Please, could you be more explicit on this point ?

I have expanded the section.

> Page 10 :
> Line 28 : Which variables do you consider for those relative errors ?

Thank you for catching this mistake; that information was indeed missing. It was the long-term average of barotropic stream function and zonally averaged temperature.

> Page 11 :
> Line 13 : It sounds that a main drawback is that Borhium an only be used in a single computational node. Have you any idea the schedule for parallelized implementation of Bohrium ?

"Automagical" distributed computing (in the sense that Bohrium automatically distributes computations between multiple compute nodes) is a hot, ongoing research problem. The developers of Bohrium have made some advances towards this [6], but a lot of work is still to be done.

However, it should be fairly straightforward to implement an abstraction to support distributed architectures at a user level, similar to the explicit style of MPI. In that case, we would have to take a step back in Veros and re-introduce some explicit parallelization logic, which is something we want to avoid as much as possible. If we can find a clean, straightforward way to support multi-node architectures without sacrificing too much flexibility, this should be achievable after a few months of work.

Page 12 :
Figure 1.
- Bh CPU and Bh GPU curves can not be distinguished.

Architecture 1 does not include a GPU benchmark, so the Bh GPU curve is not present in this panel. While Bh CPU and Bh GPU performance are mostly identical on architecture 2, they differ for large problem sizes, which is the only intended take-away message of the GPU curve.

- I do not clearly understand what does the Âń line fit Âż for the MPI curve ?

This annotation indicates that *all* solid lines are line fits (in the sense of 'y = mx + b') to the available data to highlight that the performance characteristics scale as expected. I updated the figure caption to make this clearer.

**3  Technical corrections**

You will find most of your comments reflected in the changed manuscript.

line 11 : it seems the character – in reference Jones et al is not necessary. (Idem page 20 line16)

By using this format, we are following the official recommendations on how to cite SciPy [7].

**Supplement:**

[revised manuscript text omitted]

---

## Referee Comment (RC2) · Anonymous Referee #2 · 21 Jun 2018

The paper presents a purely python based ocean model (VEROS), which is translated from the the pyOM2 model with a fortran backbone. The author addresses many of the implementation obstacles that had to be solved in python to obtain a python based ocean model with a descent performance. In general the author did a good job in presenting their way of translating and implementing VEROS. To my knowledge, is the introduction of a purely python based ocean model, mainly for teaching purposes and as an easy entry for beginners as well as a fast testing tool for new ideas, a novel effort in the ocean model community. I would therefor recommend that the paper is accepted after some minor revision.

General Comments: -Introduction,line 6: ...using the low level programming language fortran... I think it should be mentioned that fortran is only one of the programming

languages used in the ocean model community, not THE language. There are plenty of models that are also written in C or C++.

-Introduction, line 8-10: ...violates a core principle of science: reproducibility... I don't see how complexity violates reproducibility. Its the job of the programmer and the development community to do implementation step by by step and that new implemented features are carefully tested, but this inherits any kind of programming effort independent of the used language.

-Introduction, line 11-12: ...designed with the explicit goal to improve code structure and readability... Any code in any language can be structured and commented that "anybody" can read and understand it, but only when the responsible programmer cares about. The difference in python to all other languages I know is, that the structure of the code (line intents, tabs, spaces...) is a necessary part of the syntax of python, which forces the programmer to structure its code to a certain extend.

-Introduction, line 16-17: ...a substantial amount of ... projects is devoted to understanding, writing, and debugging legacy Fortran code... Understanding writing and debugging is part of any kind of model programming effort, irrespective of the used language, that is a burden one always has to deal with. I think the big advantage of the high level programming language python is, that its first unless like MATLAB fully open source, so there are no nasty licensing issues to address for any package and second that the running of the code and the visualization of any kind of model variable can be done theoretically together. This would make it much easier, especially for beginners, to understand what is going on in the model and speed up any debugging work-flow considerably. Low level programming languages only allow limited output to the screen/log-file or need own complicated output routines to write out more complex variables which are visualized with something afterwards which makes it often time and resources consuming to find the origin of bugs.

- 2.1 From Fortran to naive python, line 28: ...arbitrary indexing in Fortran... I'm not

sure what the author means here with the term "arbitrary indexing". Also indexing in Fortran is anything else than arbitrary.

- 2.3.3 Multi-threaded I/O with Compression, line 27: . . .ranging from Gigabytes to Petabytes... I haven't met yet any model application where single output files in the size of Petabyte where written. Did the author meant Terabyte ?

- 2.3.3 Multi-threaded I/O with Compression, line 29-30: Since writing output becomes more and more to a critical bottleneck especially, for large model configurations, it would be nice if the author could describe a bit more in detail how writing the output is organized in VEROS especially with respect to the separated threads. How is prevented that the data are overwritten when the model runs further, while one thread is writing out ?, Are the output data duplicated for writing the output?, Does it affect the RAM demand of the model? . . .

-2.3.5 Modular Diagnostic Interface: How does VEROS structure the output, does it follow the CMIP protocol, one file for one variable or it combine several variables into a file?

-4.1. Modified Geometry with flexible Resolution: The author should mention what he used as forcing to obtain these results

It would be nice if the author could also make some statements about the memory (RAM) demand between VEROS and pyOM2, when running the same configuration. Are they the same?, Are there differences in the size of the model configuration that VEROS can handle compared to pyOM2...

Technical Comments: - page1, line15: ...to further advance our. . .

- page3, line26 (same page8, line9): 2.1 From Fortran to naïve Python

- page9, line24: ...and the implementation of

- page16, line21: ...(Chelton and coauthors et al. ,1998)

- page16, line25: ...(Chelton and coauthors et al. ,1998)

- page19, line 34: . . . Last Glacial Meaximum. . .

- page 21,line 4: ...PopularitYy...

Please also note the supplement to this comment:
https://www.geosci-model-dev-discuss.net/gmd-2018-3/gmd-2018-3-RC2-
supplement.pdf

---

## Author Comment (AC2) · 1 Jul 2018

Dear referee,

thank you for your thoughtful review and the recommendation to accept our paper. We address the issues you raised point-by-point below. You will also find a `latexdiff` of the revised document as a supplement.

Kind regards,

Dion Häfner
on behalf of the authors

[Figure]

**1 General Comments**

-Introduction,line 6: ...using the low level programming language fortran ... I think it should be mentioned that fortran is only one of the programming languages used in the ocean model community, not THE language. There are plenty of models that are also written in C or C++.

Re-worded these sentences to be a bit more general.

-Introduction, line 8-10: ...violates a core principle of science: reproducibility ... I don't see how complexity violates reproducibility. Its the job of the programmer and the development community to do implementation step by by step and that new implemented features are carefully tested, but this inherits any kind of programming effort independent of the used language.

Complexity per se indeed does not automatically *violate* reproducibility. However, we live in a world where resources are finite, and especially so in academia. A more complex model code that is poorly abstracted *does* lead to a decrease in "testability", and since people's time is precious, this particular piece of code will be, on average, less well tested.

Changed the wording of this sentence ("violate" → "jeopardize").

-Introduction, line 11-12: ...designed with the explicit goal to improve code structure and readability ... Any code in any language can be structured and commented that "anybody" can read and understand it, but only when the responsible programmer cares about. The difference in python to all other languages I know is, that the structure of the code (line intents, tabs, spaces ... ) is a necessary part of the syntax of python, which forces the programmer to structure its code to a certain extend.
Our understanding of structure in a program goes far beyond whitespace. Python (such as many other high-level languages) actively advocates the usage of modularity, clear scoping rules, and other modern software engineering best practices. Furthermore, the community-wide coding standard PEP8[1] demands a consistent style of all Python projects, lowering the bar of entry for new collaborators.

Please also consider the following quote from our response to the first review comment:

"While it is in principle possible to write clear Fortran code with meaningful abstractions that may be just as readable as a high-level implementation, the reality is often different. Popular ocean models such as MOM [4] or POP2 [5] feature subroutines that are hundreds to thousands of lines long, and both models rely on more obscure Fortran features such as `COMMON` blocks, which makes it hard to keep track of variable scopes for inexperienced programmers. This is not necessarily due to flaws in Fortran's core design, but we do consider the established idiomatic style of a community to be tightly bound to the language used."

-Introduction, line 16-17: ... a substantial amount of ... projects is devoted to understanding, writing, and debugging legacy Fortran code... Understanding writing and debugging is part of any kind of model programming effort, irrespective of the used language, that is a burden one always has to deal with. I think the big advantage of the high level programming language python is, that its first unless like MATLAB fully open source, so there are no nasty licensing issues to address for any package and second that the running of the code and the visualization of any kind of model variable can be done theoretically together. This would make it much easier, especially for beginners, to understand what is going on in the model and speed up any debugging work-flow considerably. Low level programming languages only allow limited output to the screen/log-file or need own complicated output routines to write out more complex variables which are visualized with

something afterwards which makes it often time and resources consuming
to find the origin of bugs.

The emphasis of this sentence is not on *understanding, writing, and debugging*, but on
*substantial amount of the duration* and *legacy Fortran code*. The debugging workflow
in a Python environment is very different from that in a Fortran project. Python code
can be pulled apart dynamically, run interactively in a Jupyter notebook, and (as you
correctly note) tightly integrated with visualization tools. On top of this, Python is one of
the most popular programming languages in the world, which means it is much easier
to get help. All of these factors contribute to a significantly more pleasant experience
for everyone who is not a Fortran expert.

On a side note, there is a multitude of stronger reasons why one would choose Python
over MATLAB than it being free software (some of which are outlined in Sect. 2.3). In
my personal opinion, a project like Veros in MATLAB would be pointless, since most of
Python's advantages don't apply.

> - 2.1 From Fortran to naive python, line 28: ...arbitrary indexing in Fortran ...
> I'm not sure what the author means here with the term "arbitrary indexing".
> Also indexing in Fortran is anything else than arbitrary.

This refers to the fact that the index range of an array can be chosen by the programmer
in Fortran [1], while Python arrays always start at 0. Slightly reworded this remark to
make that clearer.

> - 2.3.3 Multi-threaded I/O with Compression, line 27: ... ranging from Giga-
> bytes to Petabytes... I haven't met yet any model application where single
> output files in the size of Petabyte where written. Did the author meant
> Terabyte ?

This sentence refers to "data sets" by which we mean the entire output of, say, a model run. In long-running high-resolution models, this easily reaches the Petabyte scale (e.g. the experiment outlined in [2]).

> - 2.3.3 Multi-threaded I/O with Compression, line 29-30: Since writing output becomes more and more to a critical bottleneck especially, for large model configurations, it would be nice if the author could describe a bit more in detail how writing the output is organized in VEROS especially with respect to the separated threads. How is pre- vented that the data are over-written when the model runs further, while one thread is writing out ?, Are the output data duplicated for writing the output?, Does it affect the RAM demand of the model? ...

Good point. All data is copied in-memory to the output thread before continuing. This indeed increases memory consumption temporarily, which is why this feature can be disabled with a flag. In practice, we have not experienced issues with excessive memory usage.

Added a sentence to Sect. 2.3.3.

> -2.3.5 Modular Diagnostic Interface: How does VEROS structure the output, does it follow the CMIP protocol, one file for one variable or it combine several variables into a file?

Currently, we loosely follow the CF conventions (`http://cfconventions.org/`) in our netCDF output files. So far, most of the output format is inherited from pyOM2, but we plan to be strictly compliant with CF in the future. For all default diagnostics, we use one file per diagnostic (e.g., all snapshot variables are written to one file, and all temporal average variables to another). But we consider this a detail that might change soon, so we don't want to get into too much detail in this paper.

-4.1. Modified Geometry with flexible Resolution: The author should mention what he used as forcing to obtain these results

Indeed. Added a sentence to 4.2.

It would be nice if the author could also make some statements about the memory (RAM) demand between VEROS and pyOM2, when running the same configuration. Are they the same?, Are there differences in the size of the model configuration that VEROS can handle compared to pyOM2...

Added a paragraph on memory consumption to Sect. 3.2.

**2   Technical Comments**

- page1, line15: ...to further advance our ...

Replaced "further" by "advance" for clarity.

- page3, line26 (same page8, line9): 2.1 From Fortran to naïve Python

Replaced the somewhat archaic spelling "naïve" with the more modern "naive".

- page9, line24: ...and the implementation of

Re-worded.

- page16, line21: ...(Chelton and coauthors et al. ,1998)

- page16, line25: ...(Chelton and coauthors et al. ,1998)

Bibliography and reference styles are as supplied by Copernicus, which we have no control over.

- page19, line 34: ... Last Glacial Meaximum ...

Corrected.

- page 21,line 4: ...PopularitYy...

We follow the official spelling of PYPL here, which is indeed "PopularitY".

**3  Other Changes**

Consistent spelling of pyOM2 with lower-case "p".

**Supplement:**

[revised manuscript text omitted]